# ApoE Isoforms Inhibit Amyloid Aggregation of Proinflammatory Protein S100A9

**DOI:** 10.3390/ijms25042114

**Published:** 2024-02-09

**Authors:** Shamasree Ghosh, Shanmugam Tamilselvi, Chloe Williams, Sanduni W. Jayaweera, Igor A. Iashchishyn, Darius Šulskis, Jonathan D. Gilthorpe, Anders Olofsson, Vytautas Smirnovas, Željko M. Svedružić, Ludmilla A. Morozova-Roche

**Affiliations:** 1Department of Medical Biochemistry and Biophysics, Umeå University, SE-90187 Umeå, Sweden; shamasree.ghosh@umu.se (S.G.); tamilselvi.shanmugam@umu.se (S.T.); igor.iashchishyn@umu.se (I.A.I.); 2Department of Medical and Translational Biology, Umeå University, SE-90187 Umeå, Sweden; chloe.williams@umu.se (C.W.); jonathan.gilthorpe@umu.se (J.D.G.); 3Department of Clinical Microbiology, Umeå University, SE-90187 Umeå, Sweden; sanduni.jayakodi@umu.se (S.W.J.); anders.olofsson@umu.se (A.O.); 4Institute of Biotechnology, Life Sciences Center, Vilnius University, LT-10257 Vilnius, Lithuania; darius.sulskis@gmc.vu.lt (D.Š.); vytautas.smirnovas@bti.vu.lt (V.S.); 5Department of Biotechnology, University of Rijeka, HR-51000 Rijeka, Croatia

**Keywords:** amyloid, S100A9, apolipoprotein E, proinflammatory, neurodegeneration, neuroinflammation, Alzheimer’s disease, cytotoxicity, fibrils, inhibition

## Abstract

Increasing evidence suggests that the calcium-binding and proinflammatory protein S100A9 is an important player in neuroinflammation-mediated Alzheimer’s disease (AD). The amyloid co-aggregation of S100A9 with amyloid-β (Aβ) is an important hallmark of this pathology. Apolipoprotein E (ApoE) is also known to be one of the important genetic risk factors of AD. ApoE primarily exists in three isoforms, ApoE2 (Cys112/Cys158), ApoE3 (Cys112/Arg158), and ApoE4 (Arg112/Arg158). Even though the difference lies in just two amino acid residues, ApoE isoforms produce differential effects on the neuroinflammation and activation of the microglial state in AD. Here, we aim to understand the effect of the ApoE isoforms on the amyloid aggregation of S100A9. We found that both ApoE3 and ApoE4 suppress the aggregation of S100A9 in a concentration-dependent manner, even at sub-stoichiometric ratios compared to S100A9. These interactions lead to a reduction in the quantity and length of S100A9 fibrils. The inhibitory effect is more pronounced if ApoE isoforms are added in the lipid-free state versus lipidated ApoE. We found that, upon prolonged incubation, S100A9 and ApoE form low molecular weight complexes with stochiometric ratios of 1:1 and 2:1, which remain stable under SDS-gel conditions. These complexes self-assemble also under the native conditions; however, their interactions are transient, as revealed by glutaraldehyde cross-linking experiments and molecular dynamics (MD) simulation. MD simulation demonstrated that the lipid-binding C-terminal domain of ApoE and the second EF-hand calcium-binding motif of S100A9 are involved in these interactions. We found that amyloids of S100A9 are cytotoxic to neuroblastoma cells, and the presence of either ApoE isoforms does not change the level of their cytotoxicity. A significant inhibitory effect produced by both ApoE isoforms on S100A9 amyloid aggregation can modulate the amyloid-neuroinflammatory cascade in AD.

## 1. Introduction

Alzheimer’s disease (AD) is a progressive neurodegenerative disorder, which is known to be the most common cause of dementia amongst the elderly population. The pathological hallmarks of AD include neuronal death, neuroinflammation, synaptic loss, extracellular protein deposits called amyloid plaques, consisting mostly of Aβ, and intracellular deposits of neurofibrillary tangles containing protein tau [1]. The formation of extracellular plaques, head injury, or infections can activate microglia and astrocytes. Activated microglia and astrocytes elicit various proinflammatory molecules such as S100 proteins, cytokines, chemokines, and reactive oxygen species, thereby causing neuroinflammation [2,3].

S100 proteins are low molecular weight, calcium-binding proteins, coordinating two calcium ions by two EF-hand structural motifs. They are multi-functional proteins reported to be involved in cell growth and differentiation, calcium homeostasis, protein phosphorylation, and other functions in the body. They mostly exist as homo- or heterodimers [3]. For example, S100A9 exists both as a homodimer as well as heterodimer with S100A8, and the latter is termed calprotectin. Both S100A9 and S100A8 are mostly expressed in neutrophils and microglia in response to inflammation and injury [4]. Studies in recent decades have shown that the S100A9 protein is a key factor in AD. It has been shown in AD mouse models that the knockdown of the S100A9 gene leads to a reduction in Aβ plaques and improves learning abilities [5]. Moreover, S100A9 possesses amyloidogenic properties and co-aggregates with Aβ in vitro, in AD, and in the traumatic brain injury (TBI) patients’ brains. These co-aggregates may serve as precursor plaques in the development of post-TBI AD and constitute senile plaques in AD [6,7,8].

ApoE has also long been associated with AD as the strongest genetic risk factor. The human *APOE* gene exists as three polymorphic alleles, *APOE* ε2, *APOE* ε3, and *APOE* ε4, which have a worldwide frequency of 8.4%, 77.9%, and 13.7%, respectively [9]. However, the occurrence of the *APOE* ε4 allele is dramatically increased to ~40% in patients with AD [9]. ApoE is a 299-residue lipoprotein that exists as three major isoforms, differing in 112/158 amino acid residue positions, viz., ApoE2^Cys/Cys^, ApoE3^Cys/Arg^, and ApoE4^Arg/Arg^ [10]. ApoE is the major lipoprotein produced in astrocytes and microglial cells in the brain for maintaining lipid homeostasis. In the brain, ApoE forms high density lipoprotein-like cholesterol-rich particles [11]. While the three isoforms differ by only one or two amino acid residues, their effects on AD are immense. ApoE4 is considered to be the strong genetic risk factor of AD, whereas ApoE3 and ApoE2 have neutral and protective effects on AD, respectively [12,13]. ApoE4 not only increases the risk, but also reduces the age of AD onset [12,14].

ApoE is known to modulate the homeostasis of Aβ, both by direct interactions [10,15,16,17] and by affecting its clearance, uptake, and degradation indirectly [18,19,20,21] in an isoform-dependent manner. In addition to its association with Aβ, the ApoE4 isoform causes neuroinflammation by promoting the release of proinflammatory mediators such as cytokines and chemokines compared to other isoforms [22,23]. ApoE4 is also found to be associated with worse outcomes in AD following TBI [24,25]. It has been shown that the expression and the secretion of both ApoE as well as S100 proteins increased post-trauma in mice [26]. ApoE isoform-dependent differences are also observed in the phagocytosis of fibrillar Aβ, which is regulated also by S100A9 [27,28]. Although it was suggested that S100A9 and ApoE may interact with each other, there is no evidence of their direct interaction available to date [29].

Therefore, the present study aims to understand the interactions between S100A9 and ApoE isoforms (ApoE3 and ApoE4) in vitro and in silico. We studied here how both lipid-free and lipidated ApoE isoforms can modulate the aggregation kinetics of S100A9. That is particularly important as ApoE is the primary lipid transporter and may be present in the form of lipoprotein particles in the brain [30]. We related the kinetic experiments with the atomic force microscopy (AFM) imaging of amyloid complexes, in order to demonstrate how the morphology of the amyloids is affected by ApoE isoforms. Moreover, by using SDS-PAGE, Western blot, and cross-linking experiments, we examined the interactions of S100A9 in the native and aggregated states with ApoE isoforms added both in lipid-free and lipidated states to shed light on specific binding sites. These studies were corroborated by MD simulation experiments. In summary, we demonstrated the significant effect that both ApoE3 and ApoE4 isoforms produced on S100A9 amyloid self-assembly, especially in their lipid-free form, and suggested the mechanisms of their interactions with S100A9, which shed further light on mechanisms leading to AD and other neurodegenerative amyloid diseases.

## 2. Results

### 2.1. Inhibition of S100A9 Amyloid Aggregation by ApoE Isoforms Monitored by Thioflavin T (ThT) Fluorescence

Figure 1A,B show the kinetics of S100A9 amyloid formation in the presence of increasing concentrations of ApoE isoforms (ApoE3 or ApoE4) monitored by the ThT fluorescence assay, as described previously [31]. S100A9 alone self-assembles into amyloids by the nucleation-dependent polymerization mechanism, and its kinetics are characterized by the lack of a significant lag phase and by a steep growth phase, prior to reaching the plateau level [32]. The addition of sub-stoichiometric ratios of ApoE3 or ApoE4, i.e., 0.31, 0.63, 1.25, 2.5, and 5 μM, compared to 50 μM S100A9, produced an inhibitory effect on the aggregation of S100A9. Upon increasing ApoE isoform concentrations, the effective growth rate decreased, as shown in Figure 1C. It is important to note that the addition of the highest concentration of ApoE3 led to a decrease in the growth rate of S100A9 aggregation by ~11% compared to the addition of ApoE4, indicating that ApoE3 is a more potent inhibitor. Interestingly, the plots of the effective growth rates versus corresponding ApoE concentrations in a logarithmic scale show two distinct slopes, indicating that two different mechanisms of ApoE inhibition of S100A9 amyloid formation are involved. The first mechanism is active in the lower concentration regime from 0.32 to 1.25 µM, and the second one dominates in the higher concentration regime from 2.5 to 5 µM, where the growth rates become slower (Figure 1C). Indeed, the oligomeric state of lipid-free ApoE is concentration-dependent [33], and some of these oligomers may be more effective in inhibiting the aggregation of S100A9, as shown in Figure 1C.

The corresponding ThT plateau levels also decreased with an increase in the concentration of either ApoE isoforms, reflecting the decrease in the overall amount of self-assembled amyloids (Figure 1A,B). To verify whether this effect is specific to ApoE isoforms rather than being the result of the addition of arbitrary proteins and their crowding effect, we examined the effects of two unrelated proteins such as bovine serum albumin and hen egg white lysozyme on the aggregation of S100A9. The data shown in Appendix A demonstrates that the influence of bovine serum albumin and lysozyme are minimal compared to that of ApoE isoforms at the same molar concentrations.

Since ApoE has a lipid-binding site in its C-terminus and forms complexes with the lipids and cholesterol [34], here we examined how the lipidation of ApoE isoforms affects the aggregation of S100A9. We reconstituted ApoE-lipid particles by using 1,2-dioleoyl-sn-glycero-3-phosphocholine (DOPC) and cholesterol, as described previously [35], denoting them ApoE-DOPC. It has been shown previously that lipidation stabilizes ApoE and prevents its aggregation [35,36]. Indeed, here we have also shown that ApoE4 is more easily cleaved by trypsin in the non-lipidated versus lipid-bound state (Appendix A). We observed that lipidated ApoE isoforms inhibited the aggregation of S100A9 in a concentration-dependent manner at concentrations from 0.25 to 4 μM compared to 50 μM S100A9, as shown in Figure 1D,E. The ThT fluorescence intensity decreased with an increase in the ApoE concentration. However, the effects of lipidated ApoE isoforms were less pronounced compared to those induced by lipid-free ApoE isoforms. The effective growth rates obtained from the half-time values of the growth phase in the amyloid formation kinetics did not change much upon increasing lipidated ApoE concentrations, as shown in Figure 1F. We found that even upon lipidation, ApoE3 produced a more noticeable inhibitory effect on S100A9 fibrillation compared to ApoE4. To examine if lipid particles themselves have any effect on the aggregation of S100A9, we monitored the aggregation kinetics of S100A9 in presence of DOPC lipid particles using ThT fluorescence. We found that the effects of the lipid particles were minimal, as shown in Appendix A. Thus, we found that both ApoE isoforms inhibit the aggregation of S100A9 in both lipidated and lipid-free states, but the effects of the latter are more pronounced.

### 2.2. AFM Analysis of S100A9 Amyloids and ApoE Isoform Aggregates

To monitor the changes in amyloid morphology, AFM imaging was performed on S100A9 samples in the absence and presence of ApoE isoforms after 15 and 70 h incubation (Figure 2A–J). S100A9 alone formed long, flexible fibrils up to a micron length after 70 h, which tended to intertwine into large tangles (Figure 2A,F and Appendix A). The height of the fibrils in the AFM cross-sections was approx. 1 nm (Appendix A). In the presence of 5 μM lipid-free ApoE3, the fibrils were short (~0.25 µm), as shown in Figure 2B,G, and Appendix A, and had a height of around 1.5 nm, as shown in Appendix A. We observed also the clumping of aggregates in certain areas, which were ~8–10 nm in height in the AFM cross-sections. These might be the aggregates of ApoE3 together with S100A9 fibrils, as their morphology was similar to ApoE aggregates alone, as shown in Appendix A. In the presence of lipid-free ApoE4, we found aggregated clusters similar to those observed for S100A9 with ApoE3 (Figure 2C,H). The fibrils of S100A9 formed in the presence of 5 μM ApoE4 had similar cross-section heights of 1.5 nm (Appendix A) to those formed in the presence of ApoE3, but the fibrils were longer compared to the ones formed in the presence of ApoE3, as shown in Appendix A. These observations also suggest that ApoE3 is a more potent inhibitor than ApoE4, which corroborates our kinetic analysis.

Upon lipidation, ApoE did not form aggregates as shown previously [35]. Figure 2D,E,I,J display the S100A9 fibrils in the presence of 4 μM lipidated ApoE isoforms at 15 and 70 h incubation, respectively. In the presence of lipidated ApoE isoforms, the fibrils formed after 70 h were long and straight. The S100A9 fibrils formed in the presence of ApoE4-DOPC were even longer and straighter than the fibrils formed in the presence of ApoE3-DOPC, as shown in Figure 2I,J and Appendix A, respectively. However, the fibril height was similar to that of S100A9 fibrils assembled in the presence of lipid-free ApoE isoforms, as shown in Appendix A. As a control, we examined the effect of only DOPC lipid particles on S100A9 amyloid aggregation. We found that there was no difference in the kinetics of amyloid formation, but the morphology of the fibrils was similar to fibrils formed in the presence of lipidated ApoE, i.e., being long and straight (Appendix A). Therefore, we concluded that ApoE isoforms modulate the morphology of S100A9 fibrils differently, when present in lipid-free or lipidated states, but the effect produced by ApoE3 and ApoE4 themselves either in their non-lipidated or lipidated forms were comparable to each other.

### 2.3. Effect of ApoE3 on S100A9 Fibrillation Phases

We observed an inhibitory effect when lipid-free ApoE3 was added at sub-stoichiometric ratio compared to S100A9 at different time points during the S100A9 fibrillation (Figure 3A). Figure 3A shows the time course of the aggregation of 50 μM S100A9 (black line) and how the addition of 5 μM ApoE3 affects the aggregation course when added at the beginning, (red line), at an early growth phase, 6 h (blue line), mid growth phase, 13 h (green line) and at the end of the growth phase, 30 h (purple line). The effect was more pronounced when ApoE3 was added at the beginning of the aggregation. When ApoE was added at various stages of the growth phase in the amyloid kinetics, it still produced an inhibitory effect reflected in the course of ThT fluorescence. We performed also AFM imaging of the amyloids at the end of the aggregation reaction. We found that the addition of ApoE at different time points was reflected in the similar morphology of S100A9 fibrils with clumps of S100A9 and ApoE, as shown in Figure 3B–E. A similar inhibitory effect was also observed with DOPC-ApoE3 added at the same time points during S100A9 amyloid kinetics, as shown in Appendix A, but the effect was less pronounced than with lipid-free ApoE.

### 2.4. ApoE Forms a Sodium Dodecyl Sulphate (SDS) Stable Complex with S100A9 during Amyloid Aggregation

To examine whether ApoE forms a complex with S100A9 during aggregation, we collected the samples of 50 µM S100A9 and 2 µM of lipidated and lipid-free ApoE isoforms after 70 h incubation and subjected them to SDS-PAGE under reducing conditions. The large aggregates did not enter the gel. Lane 1 in Figure 4A shows only monomeric and dimeric S100A9 corresponding to 13 and 26 kDa bands, respectively. There is also a faint band corresponding to the trimer of S100A9 (~39 kDa). The next four lanes correspond to species of S100A9 in the presence of 2 μM ApoE isoforms in their lipidated as well as lipid-free states. The band near 37 kDa corresponds to ApoE isoforms (~34 kDa). The band marked at ~50 kDa might be the SDS stable complex of S100A9 with the corresponding ApoE isoform. This might be a complex of S100A9 with ApoE at the molar ratio of 1:1. To confirm this, we performed a Western blot using polyclonal anti-S100A9 antibodies, and we observed the same band at ~50 kDa, as shown in Figure 4B; however, we did not observe any band of S100A9 alone at this position, when only S100A9 aggregates were added to the gel (Lane 1, Figure 4B). Therefore, this confirmed that this band corresponds to the ApoE and S100A9 complex with ~50 kDa molecular mass. In the gel with Coomassie blue staining, this band was very faint, as shown in Figure 4A. However, in the Western blot with the anti-S100A9 antibody, we saw a bright band (Figure 4B). The low sensitivity of Coomassie blue staining [37] and the high sensitivity of antibodies in the Western blot can explain such an observation. The S100A9–ApoE complex band intensities were stronger for lipid-free ApoE isoforms compared to lipidated ApoE isoforms. This suggests that the C-terminal of ApoE, corresponding to the lipid-binding region [30], might be the main binding region with S100A9. This result is further consistent with our kinetics data presented in Figure 1, showing that lipid-free ApoE isoforms are more potent inhibitors compared to lipidated ApoE isoforms. We also observed a band of ~60 kDa in the Western blot lanes corresponding to S100A9 and ApoE isoform complexes in Figure 4B. This band may correspond to a complex of S100A9 dimers and ApoE monomers (~60 kDa) or S100A9 tetramers (~53 kDa). However, we did not observe this band when only S100A9 aggregates were loaded into the gel. Therefore, we suggest that ApoE forms a complex with the dimer of S100A9, giving rise to a band of ~60 kDa. We observed also some higher molecular weight bands of ~70 kDa in SDS-PAGE lanes, as shown in Figure 4A, corresponding to S100A9 aggregated with ApoE isoforms. These bands correspond to the dimer of ApoE as they appeared in the lanes, which contain only ApoE aggregates (Figure 4A).

In order to examine further the interactions of S100A9 with ApoE isoforms during aggregation, we aggregated 100 μM S100A9 and 5 µM lipid-free ApoE4 for 70 h and centrifuged the sample to obtain the pellet and supernatant (Figure 4C,D). The supernatant was then subjected to size exclusion chromatography followed by SDS-PAGE. The protocol is shown in Appendix A. As shown in Figure 4C, when only the supernatant of S100A9 was subjected to the Superdex 200 column, we observed two major peaks. One came in the void volume (Peak 4), and the second came at the position of native S100A9, i.e., at 15 mL (Peak 5). When the pellet, as well as the gel filtration fractions, were subjected to Western blot, two bands corresponding to 13 kDa monomer and the 26 kDa dimer of S100A9 (Figure 4D, Lanes 1, 4 and 5, respectively) were observed. In the presence of ApoE4, we found two extra peaks in the 11 and 12 mL fractions from the gel filtration column (Peaks 7 and 8), respectively, apart from a peak in void volume (Peak 6) and the 15 mL fraction (Peak 9), which corresponded to larger particles and native S100A9, respectively. In the Western blot analysis, we observed the bright band of the ~50 kDa S100A9–ApoE complex in the pellet fraction (Lane 2) and the corresponding faint band in the void volume (Lane 6) in Figure 4D. We did not observe this band in other samples due to their larger dilution.

We wish to note that we used two different polyclonal S100A9 antibodies in our experiments. In Figure 4B, we showed the Western blot subjected to the Invitrogen polyclonal antibodies, and we found that this antibody was more sensitive towards the dimer of S100A9. Thus, we found a faint monomer band in this blot. On the other hand, when we used the Santa Cruz S100A9 polyclonal antibodies as shown in the Western blot in Figure 4D, which were more sensitive towards the monomeric form of S100A9, we observed a thick S100A9 monomer band in that blot.

### 2.5. Transient Interactions between Native S100A9 and ApoE Isoforms

To investigate if native S100A9 and ApoE interact with each other prior to the amyloid formation, we performed cross-linking experiments using glutaraldehyde, which were followed by SDS-PAGE and Western blot analysis. The optimal incubation time with glutaraldehyde was 30 min, when the faint band of the S100A9–ApoE cross-linked complex (~50 kDa) appeared in SDS-PAGE. In the Western blot with Invitrogen polyclonal S100A9 antibodies (Figure 5A), we observed an intense band of ~26 kDa corresponding to the S100A9 dimer, when no glutaraldehyde was added. Its intensity increased further with the increase in glutaraldehyde concentration. In the Western blot (Figure 5D) with the Santa Cruz polyclonal S100A9 antibodies, we also observed a gradual increase in the intensity of the ~26 kDa band, corresponding to the S100A9 dimer with an increase in glutaraldehyde concentration. Here, we used different antibodies as mentioned above; however, the Coomassie blue staining shows a similar trend in the increasing intensities of the ~26 kDa dimer bands upon the increasing concentrations of glutaraldehyde (Appendix A).

When equimolar concentrations of ApoE and S100A9 were mixed, both ApoE isoforms, viz., ApoE3 and ApoE4 in their lipid-free states form a complex with S100A9, which is manifested in the ~50 kDa band in the Western blots, as shown by the arrows in Figure 5B,E. The intensity of the complex band increased with the increase in the concentration of the cross-linker, with maximal intensity observed at 100X glutaraldehyde. We observed also the ~60 kDa band in the same Western blots, when 100X glutaraldehyde was added, which likely corresponded to a complex of ApoE with dimeric S100A9 (Figure 5B,E). Since the noticeable bands in the Western blots were observed upon increasing concentrations of the cross-linker, the interactions between S100A9 and the corresponding ApoE isoform are transient. To summarize the above results, we have shown the schematic presentation of the possible combinations of molecular masses upon S100A9 and ApoE oligomerization and S100A9–ApoE complex formation, which match the complexes observed in the cross-linking, SDS-PAGE, and Western blot experiments (Appendix A). The combination of molecular masses is limited, and no additional bands of unknown nature were observed, further validating the above results.

Moreover, upon the lipidation of ApoE, the interactions with S100A9 became significantly weaker, and even at 100X glutaraldehyde the intensity of the Western blot band of ~50 kDa was very faint for both ApoE3 and ApoE4 complexes with S100A9 (Figure 5C,F).

### 2.6. MD Simulation of ApoE and S100A9 Interactions

To further investigate which amino acid residues of ApoE3 and ApoE4 interact with S100A9, we performed molecular docking and MD simulation. The structures and interactions of the S100A9 dimer with the ApoE4 monomer were described by using a combination of the molecular docking based on a coarse-grained simulation for 20 μs (Appendix A) and an all-atom MD simulation for 300 ns (Figure 6) [38]. Molecular docking was used for the initial search of the binding sites of ApoE4 to the S100A9 dimer (Appendix A) [39]. Both proteins are compact without significant cavities or crevices on their surfaces. ApoE4 has relatively loose loops at the N- and C-termini, which form initial contacts with the S100A9 dimer. MD analysis revealed that ApoE4 has a diagonal ridge with positively charged Arg and Lys residues, including R92, R136, R226, R228, R251, K282, and H299, while S100A9 has a negatively charged diagonal ridge, including residues D65A, D71A, E77A, and E78A of the monomer denoted as A, and D65B, D67B, E77B, E78B, and E11B of the S100A9 monomer denoted as B (Figure 7). These amino acid residues provide specific donor–acceptor interactions within the complex. It is interesting to note that the dimer of S100A9 shows symmetrical projections involved in the complex with the ApoE4 monomer. The donor–acceptor bonds between the monomer ApoE4 and the dimer S100A9 are summarized in Appendix A. Among these pairs, nine donor–acceptor pairs showed above 60% occupancy, indicating that in more than 60% of the frames calculated in MD, these molecular bonds were present (Appendix A). At the same time, 27 bonds show above 20% occupancy. These indicate that the interactions are dynamic. Most of the Asp and Glu amino acid residues of S100A9 involved in the complex formation with ApoE are in its second EF-hand calcium-binding site (Appendix A). Most of the amino acid residues of ApoE involved in the interactions are located in its C-terminus. Since the lipidation of ApoE takes place in its C-terminus, including the amino acid residues of 244–272 [40], this may explain why the lipidated states of ApoE are less prone to form complexes with S100A9.

Similar MD calculations were performed for the interactions of the ApoE3 monomer and the S100A9 dimer, which led to similar results. ApoE3 differs from ApoE4 in its Cys112 to Arg112 substitution. This residue is not located on the interaction surface; therefore, its effect on protein–protein interactions is very subtle.

### 2.7. ApoE Isoforms Do Not Mitigate the Cytotoxic Effect of S100A9 Amyloids on Neuroblastoma Cells

To investigate whether ApoE isoforms can mitigate the cytotoxic effect of S100A9 aggregates on SH-SY5Y neuroblastoma cells, we used the MTT assay to measure cell viability. The MTT assay provides a measure of cell viability based on cellular NAD(P)H-dependent oxidoreductase enzyme activity [41]. Cells were treated for 24, 48, and 72 h in a pilot MTT assay. Changes in viability remained constant over 48 and 72 h, and here we present the results after the 24 h incubation of native and aggregated proteins with neuroblastoma cells (Figure 8). When native S100A9 was added to cells, their viability decreased by about 20%. This was observed previously by Wang et al. [7], when 5–20 µM of S100A9 was used. Here, we used 10 µM of native S100A9. It is widely recognized that S100A9 can lead to an inflammatory response and that neuroinflammation is a primary driving event in AD; therefore, S100A9 may exert cytotoxicity in its native state. The viability of the cells reduced further by 40 and 50%, when treated with 5 μM and 10 μM of aggregated S100A9, respectively (Figure 8 and Appendix A). A concentration-dependent decrease in cell viability was observed at 2.5 to 10 μM concentrations of added S100A9 amyloids, though there were no statistically significant changes in cell viability, when S100A9 amyloids were added at concentrations 1.25 μM and below (Appendix A). It was not feasible to test S100A9 concentrations greater than 10 μM, due to the potential osmotic effects of adding a high amount of proteins to cells. Interestingly, ApoE isoforms themselves were also toxic, as shown in Appendix A. The presence of ApoE alone reduced cell viability by 20% both in the native as well as in aggregated states.

Importantly, the presence of ApoE isoforms in their lipid-free as well as lipidated states did not change the toxic effect of the S100A9 amyloids (Figure 8). Therefore, we concluded that the presence of ApoE isoforms both in lipid-free as well as in lipidated states does not mitigate the cytotoxic effect of S100A9 amyloids. In order to confirm further that ApoE isoforms do not mitigate S100A9 amyloid cytotoxicity, we used Calcein-AM staining in addition to the MTT assay. Calcein-AM staining provides a measure of the number of live cells based on cellular of esterase enzyme activity [42]. Results from Calcein-AM staining support our findings from the MTT assay, specifically that the presence of lipidated ApoE isoforms did not change the toxic effect of S100A9 amyloids on neuroblastoma cells (Appendix A).

## 3. Discussion

ApoE produces pleiotropic effects in various human diseases, from AD to atherosclerosis. Over the past two decades, in numerous in vitro and in vivo experiments, it has been shown that ApoE isoforms interact with various amyloidogenic proteins such as Aβ, tau, α-synuclein, and prion proteins [43]. ApoE isoforms produce differential effects on systemic inflammation as well as neuroinflammation. ApoE4 is associated with increased levels of proinflammatory cytokines such as IL-8 and TNFα after cardiopulmonary bypass surgery [44]. It is also linked to inflammation-related metabolic disorders such as diet-induced adipose tissue inflammation. ApoE4 is even related to the worst outcomes after TBI, causing microglial activation [45], prolonged coma [46], or increased risk of AD [24]. The synergistic effects of TBI consequences and ApoE4 expression in patients with AD can be achieved via an altered state of inflammation [24]. However, the specific link between ApoE isoforms and immune cell response activation involved in AD neurodegeneration remains unsolved.

S100A9 was also used as a marker of inflammation for various diseases, including inflammation bowel disease [47], rheumatoid arthritis [48] and AD [49]. It is also shown to aggregate together with Aβ in vitro as well as in vivo in the AD and TBI patients’ brains [4,6,8]. Even though there is no direct evidence of interactions between S100A9 and ApoE isoforms reported in the literature, there is a possibility that interactions between these proteins exist, as their expression is linked to various common diseases. Therefore, here we examined how ApoE isoforms interfere with the aggregation process of S100A9 and whether they affect the toxicity of S100A9 aggregates towards neuronal cells.

We have selected ApoE isoforms ApoE3 and ApoE4 for comparison, because they differ by only one amino acid substitution at position 112, Cys112 in ApoE3 and Arg112 in ApoE4, yet, ApoE4 is viewed as the genetic risk factor of AD, while ApoE3 is neutral towards AD [10,13]. Here, we have shown that both ApoE isoforms inhibit the aggregation kinetics of the S100A9 (Figure 1A,B) at sub-stoichiometric amounts. We have observed that under fixed S100A9 concentrations, the slope on the double logarithmic plot of reaction rate versus concentration decreases upon an increase in ApoE concentrations, indicating its effect on the rate-limiting step of the aggregation (Figure 1C). Any type of aggregation goes through a rate-limiting step, which, for S100A9, is the self-assembly of two native homodimers into a tetramer [32,50]. In addition, the mechanisms of the secondary nucleation and fragmentation were shown not to be involved in S100A9 amyloid formation [6]. Therefore, ApoE isoforms affect the aggregation of S100A9 via interactions with the rate-limiting species.

Moreover, the effects on the S100A9 amyloid aggregation of both ApoE isoforms were similar, because the slopes of the concentration-dependence of the effective rates are the same; therefore, both isoforms interact with the critical nuclei in the same manner. Though the effect of ApoE3 was somehow more pronounced compared to ApoE4, the difference in the effective aggregation rates did not exceed ca. 10%. Since ApoE3 and ApoE4 differ from each other by only one amino acid residue, and that this residue is not involved in S100A9–ApoE interactions as shown by MD analysis, this deviation did not produce a significant effect on S100A9 amyloid self-assembly.

We examined also the effect on S100A9 amyloid aggregation of lipidated ApoE isoforms, since under physiological conditions lipidation stabilizes ApoE isoforms, preventing them from degradation [30,51]. Here, we also showed that ApoE4 is more easily cleaved by trypsin in the lipid-free compared to lipid-bound state (Appendix A). We found that upon the lipidation of ApoE, its effect on the S100A9 aggregation significantly decreased (Figure 1). It has been shown previously that the principal lipoprotein-binding sites of ApoE lie within the amino acid residue sequence of its C-terminal domain [22]. We have shown by using MD simulation that S100A9 interacts with ApoE4 and ApoE3 isoforms via the diagonal ridge with positively charged Arg and Lys residues from its C-terminal domain, i.e., R136, R226, R228, R251, K282, and H299. Therefore, if these residues are involved in lipid binding, this area becomes inaccessible for complex formation between S100A9 and ApoE.

We found that when lipidated ApoE was added to S100A9, the S100A9 fibrils became much longer and straighter (Figure 2I,G). A similar effect on the morphology of S100A9 fibrils was also noticed in the presence of only lipids (Appendix A). Therefore, this effect might be because of the lipid composition and physical properties of lipids [52] as well as ApoE itself. More studies are required to find out the mechanisms of the effects of lipids on the aggregation of the S100A9 protein.

Since ApoE is likely to affect the critical nuclei of S100A9 on its path to the amyloid self-assembly discussed above, we studied also the S100A9 and ApoE complex formation by using the combination of SDS-PAGE, Western blot, cross-linking experiments, and MD simulation (Figure 4, Figure 5, Figure 6 and Figure 7). All the above mentioned techniques have shown the direct interactions between proteins. We observed S100A9–ApoE complexes with stoichiometric 1:1 and 2:1 ratios both under native conditions and after 72 h of aggregation.

Interestingly, upon the mixing of native proteins, S100A9–ApoE complexes with both 1:1 and 2:1 ratios were observed in the Western blots only upon the introduction of a significant 100X excess of glutaraldehyde cross-linker (Figure 5B). This indicates that the interactions between S100A9 and ApoE in solution are highly dynamic and transient. The Western blot band corresponding to the 1:1 complex of S100A9 to ApoE is brighter and thicker than that of the 2:1 complex, indicating that the former is predominantly populated. After prolonged incubation for 72 h at 42 °C, both S100A9–ApoE complexes were also formed, and they were SDS-stable without adding the cross-linker (Figure 4). Indeed, complex formation is an activation process; therefore, prolonged incubation at 42 °C produced higher yields and stable outcomes for both the 1:1 and 2:1 stoichiometries of S100A9 and ApoE complexes than mixing the native proteins in the cross-linking experiment and MD simulation. Thus, both in vitro experiments and in silico modeling showed the significance of S100A9–ApoE complexes in inhibiting S100A9 amyloid aggregation.

Importantly, we did not find the mitigating effect of ApoE isoforms both in lipid-free and lipidated forms on the cytotoxicity of S100A9 aggregates towards neuronal cells by using two cell viability assays such as MTT and Calcein-AM (Figure 8). We have shown previously that S100A9 is a key player in the amyloid-neuroinflammatory cascade together with Aβ peptide, which is a hallmark of AD [7]. Both S100A9 and ApoE isoforms interact and co-aggregate with Aβ peptide, as demonstrated previously [7,8,10,53]. The aggregates consisting of S100A9 and ApoE isoforms might alter the Aβ aggregation differently than each protein alone, thereby modulating differently the amyloid-neuroinflammatory cascade in AD. Moreover, S100A9 is also a ligand to receptors for advanced glycation end products (RAGE) and Toll-like receptors 4 (TLR 4) in microglia [7]. The interactions of S100A9 with these receptors induce the secretion of proinflammatory cytokines by microglia, thereby causing and exacerbating neuroinflammation [7,54]. We have shown previously that if the level of S100A9 is sustained during prolonged neuroinflammation, this may lead even to the development of AD after TBI [8]. Therefore, it will be important to explore this avenue in the future and to clarify whether S100A9 interactions with ApoE isoforms in both lipid-free and lipidated states can interfere with the receptor signaling pathways in microglia cells, and thereby affect neuroinflammation.

In summary, we have shown that both ApoE3 and ApoE4 isoforms in lipid-free and lipidated states inhibit the amyloid formation of S100A9 protein. Since both isoforms differ only by one amino acid residue, we have not observed significant differences in their inhibitory effects on S100A9 aggregation. Importantly, the lipid-free ApoE isoforms were much more efficient in inhibiting S100A9 amyloid formation. We have shown that S100A9 can form transient complexes with ApoE isoforms in lipid-free states at 1:1 and 2:1 stoichiometries. These interactions involve the Asp and Glu residues in the C-terminal EF-hand calcium-binding site of S100A9 and Arg and Lys residues located in the C-terminal domain of ApoE isoforms. Since the lipid-binding site of ApoE isoforms is located at their C-termini, the lipidation of ApoE isoforms hinders their interactions and complex formation with S100A9. Consequently, the lipidation of ApoE isoforms hinder also the inhibitory effect of ApoE on the S100A9 amyloid formation, as mentioned above.

We have suggested that the inhibitory effect of ApoE isoforms on S100A9 aggregation is achieved via their complex formation, and though the complexes are transient, they can be stabilized upon prolonged co-incubation during the aggregation process. Since the critical nuclei involved in S100A9 amyloid formation are present at very low concentrations, the inhibitory effect of ApoE isoforms is pronounced even at their sub-stoichiometric ratios. Both S100A9 and ApoE interact and co-aggregate with Aβ peptide, driving the amyloid-neuroinflammatory cascade in AD; therefore, those interactions may contribute to the cascade as a whole and modulate the development of AD pathology.

## 4. Materials and Methods

### 4.1. Preparation of Lipid-Free and Lipidated ApoE

Recombinant ApoE3 and ApoE4 were obtained from AlexoTech AB, Umeå, Sweden. Lyophilized powder was dissolved in 20 mM NaOH prepared in phosphate-buffered saline (PBS) buffer and subjected to size-exclusion chromatography using Superdex 200 10/300 GL (GE Life Science, Uppsala, Sweden) in PBS buffer containing 1 mM NaN_3_. PBS solution was prepared by dissolving PBS tablets (Medicago, Uppsala, Sweden) in distilled water. Lipidated full-length ApoE isoforms were prepared using the protocol described earlier [35]. DOPC (Avanti Polar Lipids) and cholesterol (Sigma-Aldrich, St. Louis, MO, USA) were mixed in a glass vial at a molar ratio of 90:5, dried under constant nitrogen gas stream, then kept overnight in a vacuum chamber to completely remove chloroform. Lipids were resuspended in PBS at a concentration of 5 μg/μL. The solution was mixed thoroughly in a vortex mixer, using intermittent procedure for 5–10 min with 1–2 min intervals, to generate liposomes. Complete hydration of liposomes was carried out by incubating the solution at room temperature for 30 min with occasional vortex mixing.

The sodium cholate dialysis method was used to obtain more homogeneous lipidated ApoE particles [55,56]. The sodium cholate at 50 mg/mL (Sigma-Aldrich, St. Louis, MO, USA) was slowly titrated into the liposome solution, using 1–2 volumes of sodium cholate for 1 volume of lipids. The turbidity of the solution cleared after 5 min of gentle vortex mixing, using 1 min intervals, and the preparation was kept at room temperature for 30–60 min. ApoE isoforms were then added to the liposome preparation, keeping ApoE:DOPC:cholesterol at the molar ratios of 1:90:5, and mixed gently for 5–10 min with 1–2 min intervals. The solution was kept at room temperature for 1 h and dialyzed using a 10 kDa cutoff membrane against PBS for 4 h at room temperature, followed by 48 h at 4 °C with changing buffer every 12 h. After dialysis, the samples were purified by gel filtration chromatography using Superdex 200 10/300 GL. The lipidation of ApoE3 and ApoE4 was confirmed by measuring the characteristic blue shift of protein tryptophan fluorescence spectrum, reduction in hydrodynamic radii of protein molecules by dynamic light scattering, and resistance to trypsin digestion as described before [17,35] (Appendix A). The concentration of ApoE isoforms was determined by absorbance measurements at 280 nm using an extinction coefficient of 44,460 M^−1^·cm^−1^. Samples were stored at 4 °C for further use.

### 4.2. Amyloid Fibril Formation

S100A9 was expressed in *Escherichia coli* and purified as described previously [57]. Lyophilized S100A9 was dissolved in PBS buffer containing 1 mM sodium azide, keeping the solution on ice. Before incubation, S100A9 samples were filtered through a 0.22 μm cut-off syringe membrane filter (Millipore Merck, Burlington, MA, USA) to remove any preformed aggregates. The concentration of S100A9 solution was determined by absorbance measurements at 280 nm using an extinction coefficient of 6990 M^−1^·cm^−1^.

The amyloid formation was carried out by incubating S100A9 in PBS buffer containing 1 mM sodium azide, pH 7.4, and 42 °C. We used the 50 μM S100A9 and 0.32; 0.63; 1.25; 2.5; and 5 μM of lipid-free ApoE isoforms during amyloid incubation. For the co-aggregation experiments with lipidated ApoE, we used 0.25; 0.5; 1; 2; and 4 μM of ApoE isoforms. In all experiments, ApoE isoforms were taken at a sub-stoichiometric ratio. As a control for the ApoE isoform effect, bovine serum albumin (Sigma-Aldrich, St. Louis, MO, USA) and hen egg white lysozyme (Sigma-Aldrich, St. Louis, MO, USA) were used in corresponding mixtures with S100A9. These proteins were weighed, dissolved in PBS buffer, and filtered with a 0.22 μm filter before use in similar procedures as described above. The concentrations of bovine serum albumin and lysozyme were determined by absorbance measurements at 280 nm using an extinction coefficients of 43,824 M^−1^·cm^−1^ and 38,990 M^−1^·cm^−1^, respectively.

### 4.3. Measurement of the Time Course of Amyloid Aggregation by ThT Fluorescence

Aggregation of S100A9 was monitored by ThT fluorescence as described previously [31]. A total of 50 μM S100A9 in the presence of ApoE isoforms was transferred into Corning 96-black-well plates with transparent bottoms (Corning, Corning, NY, USA). Each sample contained 20 μM ThT. The sample volume was kept at 100 µL per well. The plates were immediately covered and placed in a plate reader (Infinite F200 PRO, Tecan, Grödig, Austria). The samples were incubated at 42 °C for 65–72 h. ThT fluorescence was recorded every 10 min interval. Filters at 430 nm and 485 nm wavelengths with a 20 nm bandwidth each were used for excitation and emission, respectively. Each protein sample was incubated in triplicate.

The effective growth rate is calculated from the half-time values of normalized amyloid growth curves using the equation
Effective growth rate (h^−1^) = 1/half-time(1)

### 4.4. Gel Electrophoresis and Western Blotting

S100A9 amyloid samples from ThT fluorescence kinetic experiments at the kinetic end time points were collected to subject them to SDS-PAGE and Western blot analyses. Samples were mixed with loading dye (reducing Laemmli sample buffer). The samples were vortexed and loaded in the 4–20% gradient gel for SDS-PAGE analysis, applying 90 V voltage for 1 h. After completing electrophoresis, separated proteins were transferred from the gel to 0.2 µm PVDF membrane (GE Healthcare, Munich, Germany) by semi-transfer method, followed by blocking with 2.5% milk powder in Tris buffer saline (TBS) overnight at 4 °C. Next, the membrane was rinsed in TBS buffer containing 0.01% Tween 20 (TBST) and incubated with S100A9 polyclonal antibody (PA1-46489, Invitrogen, Carlsbad, CA, USA and SC-20173, Santa Cruz, CA, USA) for 1 h. After washing with TBST buffer three times, the membrane was incubated with anti-rabbit IgGs conjugated with horseradish peroxidase (AS10668, Agrisera, Umeå, Sweden) for 1 h and washed with TBST three times again. The immunoblot was developed with a Western blot detection reagent (ECL bright, Agrisera, Umeå, Sweden), and the bands were observed on ChemiDoc MP imaging system (Bio-Rad, Laboratories, Hercules, CA, USA).

To evaluate further which state of S100A9 forms a complex with ApoE isoforms, 100 μM S100A9 in the presence of 5 μM ApoE4 was incubated at 42 °C for 72 h, followed by centrifugation at 14,000 rpm for 15 min. The supernatant was collected and subjected to the Superdex 200 column. The pellet was re-suspended in an equal volume of PBS buffer. The pellets and each fraction of elution peaks were collected and subjected to SDS-PAGE and Western blot as described above.

### 4.5. Chemical Cross-linking of S100A9 and ApoE Isoforms

To capture the complex formation between S100A9 and ApoE isoforms in the native state, chemical cross-linking between them was performed followed by SDS-PAGE and Western blot. Glutaraldehyde (Sigma-Aldrich, St. Louis, MO, USA) was dissolved in water at a concentration of 5 mM. Gel filtration purified ApoE isoforms (lipidated and lipid-free) and the S100A9 was incubated at the molar ratio of 1:1 at 4 °C overnight. A total of 4 μM of ApoE isoforms and S100A9 were used. Superdex 200 column was used to purify both ApoE as well as S100A9. The cross-linking reaction was performed by adding 10- and 100-time excess molar ratio of glutaraldehyde to the preincubated ApoE isoform and S100A9 mixtures. The reaction mixture containing proteins and glutaraldehyde was incubated for 30 min at 4 °C. Cross-linking was quenched by the addition of 1.5 M Tris (Sigma-Aldrich, St. Louis, MO, USA) to a final concentration of 50 mM for 10 min at room temperature. These samples were subjected to SDS-PAGE and Western blot experiments as described above.

### 4.6. AFM Imaging

AFM imaging was performed by using a BioScope Catalyst AFM (Bruker, Billerica, MA, USA), operating in peak force mode in the air. The scan rate was 0.51 Hz, and the resolution was 512 × 512 pixels. Bruker MSLN and SLN cantilevers were used in all measurements. We also used a Multimode AFM with a Nanoscope IV controller for a few of our samples. In these experiments, we used standard AFM cantilevers (Mikro Masch, Sofia, Bulgaria). Samples were diluted 10 times in MilliQ water before sample preparation. In total, 10 µL of each sample was deposited on the surface of freshly cleaved mica, kept for 30 min, washed five times with 200 µL of deionized water, and left to dry overnight at room temperature. Heights of amyloid fibrils were measured in the AFM cross-sections by using Bruker Nanoscope analysis software (version 1.9) or Gwyddion 2.62 [58]. The length of the fibrils was measured by using the Fiji version of ImageJ image analysis software (version 2.9.0) [59].

### 4.7. Molecular Docking Studies

The S100A9 dimer structure used in this study was the average from a set of ten NMR structures downloaded from the protein data bank, ref. PDB: 5I8N [60]. The ligands were hydrogenated and charged at pH 7.2, using the Gasteiger protocol [39]. The structure of human ApoE3 was based on the NMR structure of full-length ApoE3 from PDB: 2L7B [61]. Since there is a structural similarity between human ApoE3 and ApoE4, the amino acid sequence alignment was performed for two proteins, and the full structure of human ApoE4 was built on homology modeling from ApoE3. S100A9 dimer and ApoE3 or ApoE4 were protonated at pH 7.2 using the AMBER98S force field [39].

Molecular docking studies started with a search box that enclosed entire proteins [39]. The docking interface started at the most flexible parts of two proteins and repeated 4 times with different initial orientations. The initial residue-based coarse-grained simulation included 500 million steps, a step size of 10 fs with a total simulation of 5 μs. Once the protein–protein complex was established, we extended the simulation to 20 μs, including 1 billion steps with 20 fs step size.

### 4.8. All-Atom MD Studies

S100A9 homodimer and S100A9 complexes with ApoE3 or ApoE4 prepared in molecular docking studies were used as starting structures for MD simulations [62]. These complexes were prepared for MD calculations using the CHARMM-GUI solution builder [63]. In a typical calculation, a protein–protein complex was placed in a water box that had about 500 thousand atoms. TIP3 models for water molecules were combined with sodium and chloride ions adjusted to a concentration of 150 mM, and the net charge was set to zero. The system relaxation used a sequence of equilibration steps at 303.15 K with Nose–Hoover coupling, and the pressure was set to 1.0 bar with semi-isotropic Parinello–Rahman coupling. The verlet cutoff scheme was combined with the LINCS constraint algorithm. System relaxation used two minimization steps that were followed by one equilibration step. The simulations analyzed molecular processes on the atomic scale for 300 ns, containing 150 million steps with the step size set to 2 fs. All simulations were run in GROMACS 2020.4 version [64] on 240 cores with 480 logical cores.

### 4.9. Cell Culture

The human neuroblastoma cell line, SH-SY5Y (ATCC CRL-2266), was purchased from ATCC and cultured in Dulbecco’s modified eagle medium supplemented with 10% fetal bovine serum, 2 mM L-glutamine and 1% penicillin/streptomycin at 37 °C in a 5% CO_2_ incubator. All cell culture reagents were purchased from Gibco, Thermo Fisher Scientific, Waltham, MA, USA, unless stated otherwise. Medium changes took place every two days and cells were split at a 1:3 ratio upon reaching 80% confluency.

### 4.10. MTT Cell Viability Assay

For the cell viability assay, 100 μM S100A9 in the presence and absence of 5 μM of lipidated and lipid-free ApoE isoforms were aggregated at 42 °C for 72 h without agitation. For the control of the native state, ApoE and S100A9, both purified by gel filtration, were mixed in the above concentration just before adding to the cells. Cells were seeded at 80,000 cells/well in flat base 96-well cell culture plates (Sarstedt, Nümbrecht, Germany) coated with 50 µg/mL poly-D-lysine (Sigma-Aldrich, St. Louis, MO, USA) in cell culture medium. After 24 h, the cells were washed twice with PBS before the addition of 100 µL reduced serum Opti-MEM medium per well (Gibco, Thermo Fisher Scientific, Waltham, MA, USA). Subsequently, 10 µL S100A9/ApoE isoform mixture, both native as well as aggregates, was added to each well, and the cells were incubated for a further 24 h. Thiazolyl blue tetrazolium bromide (MTT, M2128, Sigma-Aldrich, St. Louis, MO, USA) was dissolved in sterile water at a concentration of 5 mg/mL and 11 µL was added per well. Cells were incubated in the dark at 37 ℃ in a 5% CO_2_ incubator for 6 h to allow for the formation of purple-black formazan crystals. The medium was then replaced with dimethyl sulfoxide (VWR) and absorbance was recorded at 570 nm using a BioTek Synergy 2 plate reader (Agilent, Santa Clara, CA, USA).

### 4.11. Statistical Analysis

The data were analyzed using double-sided t-test. *p*-values less than 0.05 were considered significant. The cytotoxicity results are presented as the mean  ±  standard deviation of 7 independent observations. The analysis was performed in Microsoft Excel 2016.

## Figures and Tables

**Figure 1 ijms-25-02114-f001:**
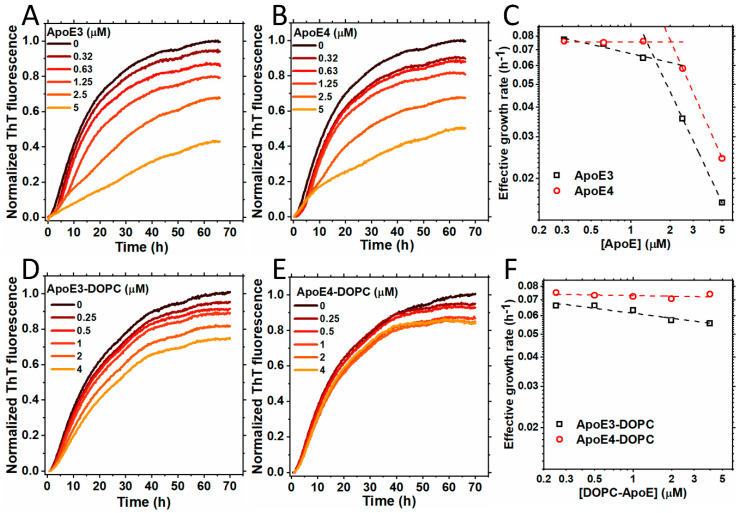
Aggregation of S100A9 in the presence of ApoE isoforms in lipid-free and lipidated forms. (**A**,**B**) Aggregation kinetics of 50 µM S100A9 in the presence of increasing concentrations of (**A**) ApoE3 and (**B**) ApoE4. The concentrations of ApoE isoforms used here are 0; 0.32; 0.63; 1.25; 2.5; and 5 μM. (**C**) The effective growth rate of ApoE3 (black dashed line and squares) and ApoE4 (red dashed line and circles) were obtained from the half-time values in (**A**,**B**), respectively. The dashed lines are to guide the eye. (**D**,**E**) Aggregation kinetics of 50 µM S100A9 in the presence of an increasing concentration of lipidated forms of ApoE isoforms, i.e., (**D**) ApoE3-DOPC and (**E**) ApoE4-DOPC, respectively. The concentrations of ApoE isoforms used here are 0; 0.25; 0.5; 1; 2; and 4 μM. (**F**) The effective growth rate of ApoE3-DOPC (black dashed line and squares) and ApoE4-DOPC (red dashed line and circles) are obtained from the half-time values in (**D**,**E**), respectively. The dashed lines are to guide the eye. The reaction is performed in PBS buffer containing 1 mM sodium azide at 42 °C in non-stirring conditions. A total of 20 μM ThT is used to monitor the aggregation kinetics.

**Figure 2 ijms-25-02114-f002:**
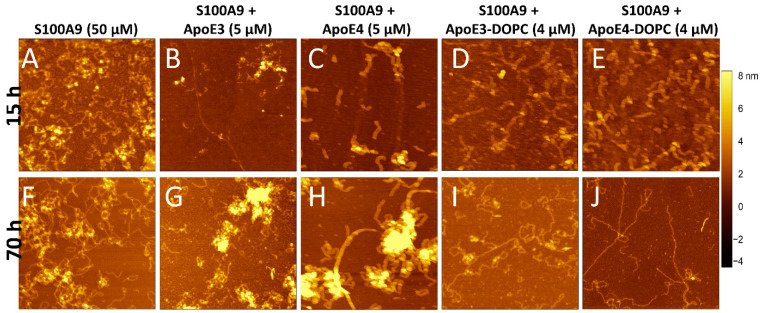
Effect of ApoE isoforms on S100A9 amyloid morphology observed by AFM imaging. AFM images of S100A9 amyloids formed (**A**,**F**) without ApoE isoforms and in the presence of (**B**,**G**) 5 µM ApoE3; (**C**,**H**) 5 µM ApoE4; (**D**,**I**) 4 µM ApoE3-DOPC; and (**E**,**J**) 4 µM ApoE4-DOPC after 15 h and 70 h incubation, respectively. A total of 50 µM S100A9 was aggregated in PBS, containing 1 mM sodium azide and in non-stirring condition at 42 °C. Scan sizes are 2 × 2 μm. The *z*-scale in AFM images is indicated by a bar with a color gradient from black to light yellow shown on the right of the images.

**Figure 3 ijms-25-02114-f003:**
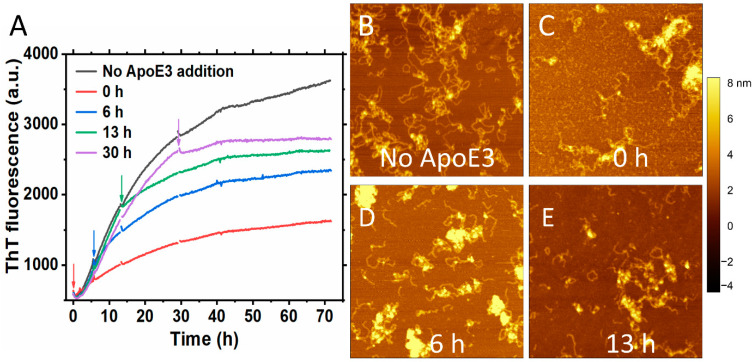
Inhibitory effect of ApoE3 on S100A9 amyloid formation persists irrespectively of the time when it is added during the growth phase of aggregation. (**A**) Aggregation kinetics of 50 μM S100A9 alone is shown in black, with the addition of 5 μM ApoE3 in the beginning of aggregation is shown in red, after 6 h aggregation is shown in blue, after 13 h – in green, and after 30 h – in purple, respectively. The arrows in the figure indicate the time points that ApoE3 was added during S100A9 amyloid formation. A total of 20 μM ThT was used to monitor the aggregation kinetics. (**B**–**E**) AFM images of S100A9 amyloids with and without ApoE3 at the end of aggregation after 70 h. (**B**) Corresponds to S100A9 aggregation without ApoE3, (**C**) ApoE was added at 0 h, (**D**)—at 6 h, and (**E**)—at 13 h, respectively. The reaction was performed in PBS buffer containing 1 mM sodium azide, pH 7.4, 42 °C, and in non-shaking conditions. Scan sizes are 2 × 2 μm. The *z*-scale in AFM images is indicated by a bar with a color gradient from black to white.

**Figure 4 ijms-25-02114-f004:**
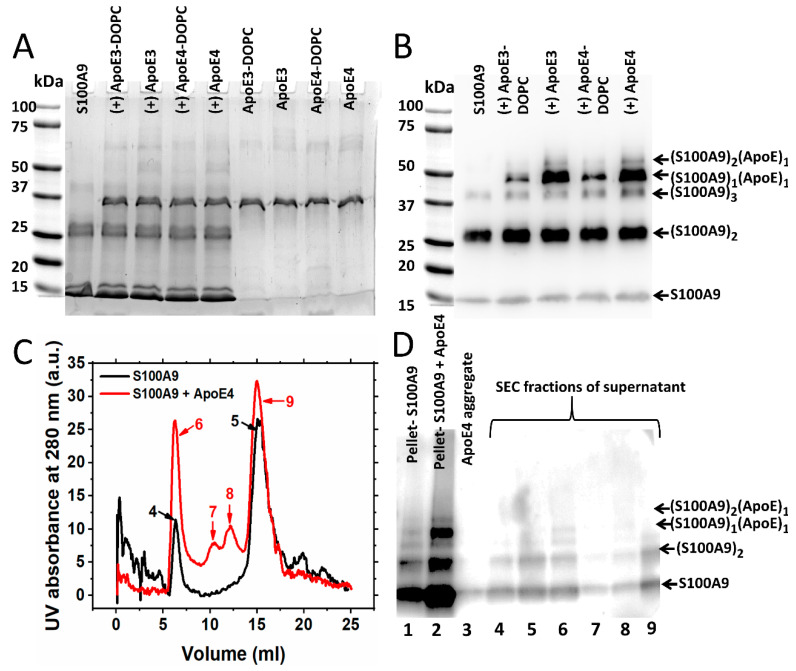
Western blot analysis of S100A9 interactions with ApoE isoforms during the amyloid formation. (**A**) SDS-PAGE of S100A9 without and with ApoE isoforms after 70 h incubation at 42 °C. Lines of SDS-PAGE indicated from left to right: S100A9 alone; S100A9 with ApoE3-DOPC; S100A9 with ApoE3; S100A9 with ApoE4-DOPC; S100A9 with ApoE4; ApoE3-DOPC alone; ApoE3 alone; ApoE4-DOPC and ApoE4 alone. (**B**) Western blot corresponding to SDS-PAGE performed using anti-S100A9 polyclonal antibody (Invitrogen, Carlsbad, CA, USA). As indicated from left to right above the Western blot: S100A9 alone; S100A9 with ApoE3-DOPC; S100A9 with ApoE3; S100A9 with ApoE4-DOPC and S100A9 with ApoE4. Bands of ~50 kDa marked in the Western blot correspond to the complexes of S100A9 with ApoE3 or ApoE4, respectively. Other bands corresponding to S100A9 and S100A9-ApoE complexes are marked on the right side from the blot. Totals of 50 μM S100A9 and 2 μM of corresponding ApoE isoforms in PBS buffer were incubated at 42 °C for 70 h. (**C**) Size exclusion chromatogram of the supernatants of S100A9 aggregated alone for 70 h (black line) and S100A9 aggregated in the presence of ApoE4 (red line) and subjected to centrifugation. The numbers of the chromatographic peaks correspond to the lane numbers in the following Western blot. (**D**) Western blot with anti-S100A9 polyclonal antibodies (Santa Cruz) of the pellets and the various peaks of supernatant, collected during elution of S100A9-aggregated samples through size exclusion chromatography. Totals of 100 μM S100A9 and 5 μM ApoE4 were incubated in PBS buffer at 42 °C for 70 h. The band with ~50 kDa molecular weight characteristic of the complex of S100A9 with ApoE4 isoform in the pellet and void volume (Lane 6) can be seen. Other bands corresponding to S100A9 and S100A9–ApoE complexes are marked on the right side of the blot.

**Figure 5 ijms-25-02114-f005:**
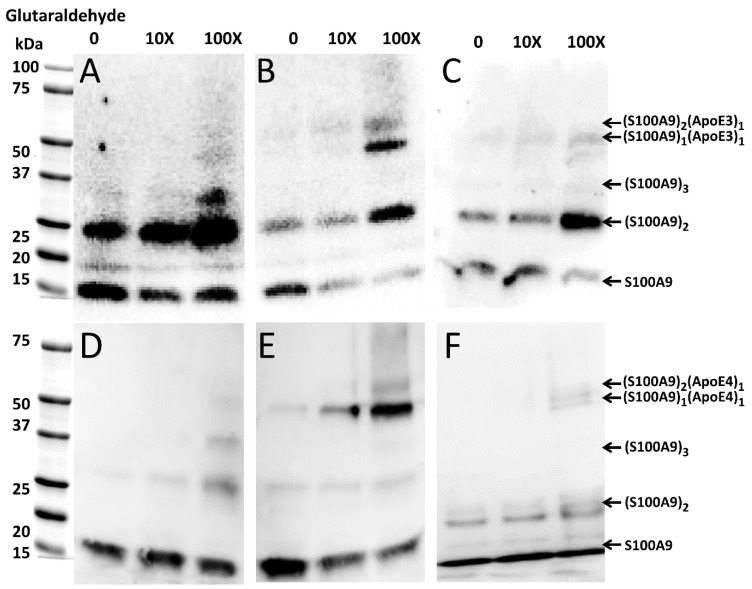
Complex formation of native S100A9 with ApoE3 and ApoE4 isoforms, respectively, observed by Western blot in the presence of increasing concentrations of glutaraldehyde. Western blot of (**A**) 4 μM S100A9 alone; (**B**) 4 μM S100A9 in the presence of 4 μM ApoE3; (**C**) 4 μM S100A9 in the presence of 4 μM ApoE3-DOPC; (**D**) 4 μM S100A9 alone; (**E**) 4 μM S100A9 in presence of 4 μM ApoE4; and (**F**) 4 μM S100A9 in presence of 4 μM ApoE4-DOPC. Western blots shown in panel (**A**–**C**) were performed by using anti-S100A9 polyclonal antibodies produced by Invitrogen, while those in panel (**D**–**F**) were performed by using anti-S100A9 antibodies from Santa Cruz. Increasing concentrations of glutaraldehyde from 0 to 10X and 100X are shown above the Western blots. The bands characteristic of S100A9 and S100A9–ApoE complexes and corresponding to all aligned blots are marked on the right side from the blots shown in (**C**,**F**).

**Figure 6 ijms-25-02114-f006:**
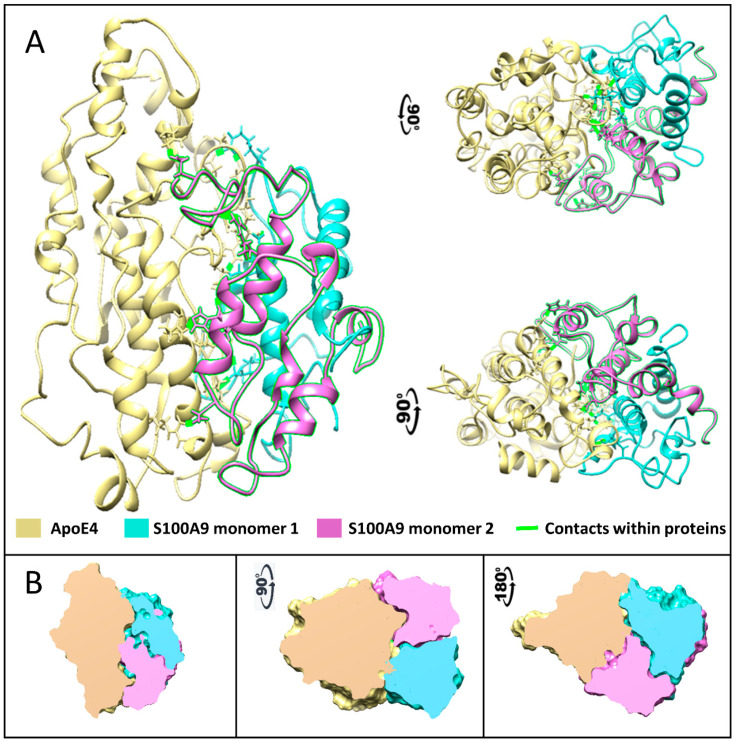
Complex formed between native S100A9 and ApoE4 calculated using multiscale MD simulation. (**A**) The protein structures are shown by ribbon backbone models in three orientations to highlight the mirror symmetry of S100A9 dimer in the complex with ApoE4. ApoE is shown in yellow, one S100A9 monomer is shown in sea blue, and another in pink. Amino acids that form contacts are shown as sticks with contacts highlighted in fluorescent green. (**B**) The cross-sections through the Connolly surfaces are used in the same three orientations to show the size and shape of protein–protein contact surfaces, illustrating large interaction surfaces and the related symmetry within the complex. ApoE is shown in yellow, one S100A9 monomer is shown in sea blue, and another in pink. The color coding of interacting proteins as in (**A**).

**Figure 7 ijms-25-02114-f007:**
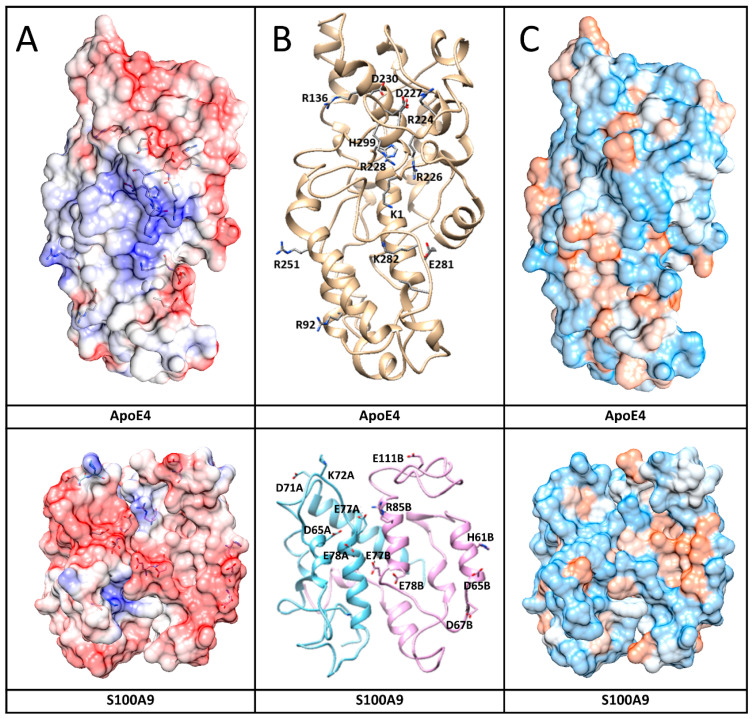
Interaction surfaces of human ApoE4 and S100A9 shown together with amino acid residues involved in the interactions. (**A**) Protein electric fields are mapped on the Connolly surface on a scale of −6 to 6 k_B_T/e, where red color coding corresponds to negatively charged residues, blue – to positive charged residues and white – to neutral residues. Interacting residues in both protein surfaces are shown by sticks. The fields clearly show a blue-positive ridge on ApoE4 that can match a red-negative ridge on S100A9. The surfaces are partially transparent to make the contact amino acids visible as sticks. (**B**) Protein backbone shows protein conformations that support formation of the complex. Sticks and related names mark the amino acids that form contacts. (**C**) Connolly surface polarity models of ApoE4 and S100A9 with polar residues shown in blue, hydrophobic – in brown, and intermediate – in white on the protein surfaces.

**Figure 8 ijms-25-02114-f008:**
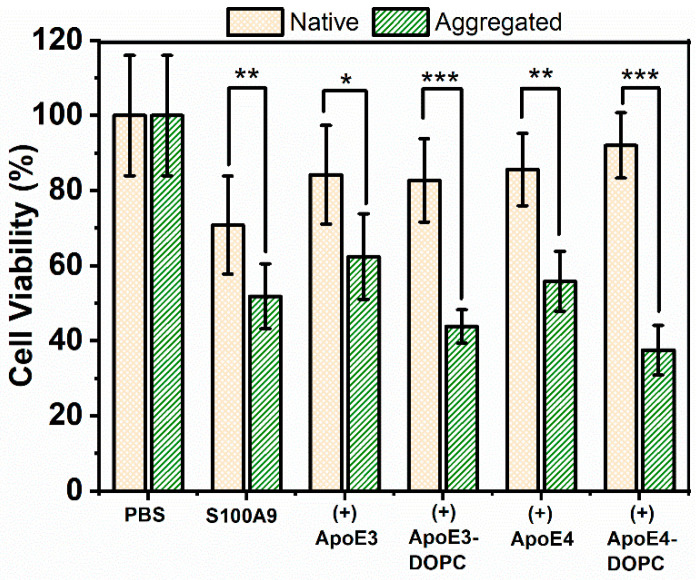
Cytotoxicity of S100A9-ApoE amyloids on SH-SY5Y neuroblastoma cells measured by the MTT assay. Viability of SH-SY5Y neuroblastoma cells was measured after 24 h co-incubation with S100A9 amyloids by monitoring the absorbance of MTT at 570 nm. Yellow bars with crossed pattern correspond to cells treated with S100A9 or S100A9 mixed with ApoE isoforms in the native state, and green bars with dash pattern correspond to cells treated with S100A9 or S100A9 mixed with ApoE in the aggregated state. The native state corresponds to freshly purified 100 μM S100A9 in the presence or absence of 5 μM ApoE mixed just prior to addition to the cells. The aggregated state corresponds to 100 μM S100A9 aggregated for 72 h at 42 °C in the presence or absence of 5 μM ApoE. Different forms of ApoE used in the experiment are shown along the *x*-axis. The final concentration of S100A9 incubated with cells was 10 μM. T-test was used to calculate *p*-values: * corresponds to *p* < 0.05, **—*p* < 0.01 and ***—*p* < 0.005.

## Data Availability

Data contained within the article.

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
