# Peer review of "ApoE Isoforms Inhibit Amyloid Aggregation of Proinflammatory Protein S100A9"

_ijms, 2024, doi:10.3390/ijms25042114_

Round 1

Reviewer 1 Report

Comments and Suggestions for Authors

S100A9 protein is a calcium-binding protein found in the amyloid plaque in human AD patients. The corresponding author of this manuscript is a leading investigator in studying the pathophysiological roles of the S100 A9, especially in neurodegenerative diseases, such as Alzheimer's disease.  This group has been taking a multifrontal approach, using rigorous biochemical and biophysical approaches to demonstrate that S100 A9, known to produce proinflammatory responses in microglia and in macrophages, can become aggregated and produce polymeric amyloid-like structures that are SDS-stable.  Here, they set out to test the hypothesis that APOE3/E4 differentially affects the S100A9 to form amyloid in vitro.  Their results demonstrate that both E3 and E4, in either a lipid-free state or in a lipid-bound state, prevent the S1900A9 from forming the amyloid in a dose-dependent manner. Molecular dynamic analysis suggests the interaction is between the C-terminal domains of APOE and S100A9 proteins.  They also showed that S100A9 is cytotoxic  to neuronal cells while the interaction of either E4 or E4 did not prevent the cytotoxicity. The last result is somewhat surprising. Perhaps the assay that they used to monitor amyloid is not the best? What if they used microglia to monitor the inflammatory response elicited by S100A9? Please address this question in the Discussion section.    

Reviewer 2 Report

Comments and Suggestions for Authors

The research carried out by the authors of the manuscript is devoted to an urgent scientific problem – the determination of the effect of the ApoE isoform on the aggregation of S100A9. The results shown by the authors of the manuscript are of unconditional interest to researchers in the field of neurobiology, biochemistry and neurology. Therefore, the conducted research is certainly relevant for publication in the journal IJMS.

The relevance and justification of the hypothesis formulated by the authors for the conducted research are given in detail and clearly in the section "Introduction". The study was carried out at a good methodological level, and the use of research methods is described in detail. The results of the research are objectively and clearly presented in the manuscript and discussed in detail in the "Discussion" section with an emphasis on current data on the direction chosen by the authors.

However, the manuscript in its current form contains the following some disadvantages:

1. It is necessary to reflect the use of statistical methods in the relevant subsection of the chapter "Materials and methods".

2. Add information about the prevalence of ApoE isoforms (if any).

Reviewer 3 Report

Comments and Suggestions for Authors

This manuscript shows an interesting study that ApoE isoforms modulate the aggregation of proinflammatory protein S100A9. Besides, ApoE and S100A9 form an SDS stable complex upon aggregation. However, a couple of questions are required to be addressed before accept.

1. Authors found that ApoE forms an SDS stable complex with S100A9. I think this complex does not exist in organisms. Thus, I wonder what is the biological and medical significance of the results? Besides, the stable complex of ApoE-SDS-S100A9 should be displayed by a structural pattern diagram. Because the current data makes it difficult to understand the mechanism of the formation of this complex.

2. The chemical crosslinking of S100A9 and ApoE isoforms only used glutaraldehyde as chemical crosslinking agent. I think that one more chemical crosslinking agents are essential, such as DSS, DSP.

3. In MTT assays, the cells were treated by S100A9-ApoE amyloids for 24 h. I think this experiment still needs to set other time points, such as 48 h and 72 h. Besides, the different doses also need.

Comments on the Quality of English Language

Please check and revise the English language thoroughly. For example, the format of SDS-PAGE is inconsistent in the full text. This symbol ‘µM’ is written wrong in some places in the text.

Round 2

Reviewer 3 Report

Comments and Suggestions for Authors

The manuscript has been revised as per the reviewer's requests and is now suitable for publication.